# The fractal structure of communities of practice: Implications for business organization

Emily Webber[1], Robin Dunbar[2]*

**1** Tacit London Ltd, Loughton, England, United Kingdom, **2** Magdalen College, Oxford, England, United Kingdom

* robin.dunbar@magd.ox.ac.uk

## Abstract

Communities of practice (COP) are informal (sometimes formal) groupings of professionals with shared interests that form to facilitate the exchange of expertise and shared learning or to function as professional support networks. We analyse a dataset on the size of COPs and show that their distribution has a fractal structure similar to that found in huntergatherer social organisation and the structure of human personal social networks. Small communities up to about 40 in size can be managed democratically, but all larger communities require a leadership team structure. We show that frequency of interaction declines as size increases, as is the case in personal social networks. This suggests that professional work-oriented organisations may be subject to the same kinds of constraint imposed on human social organisation by the social brain. We discuss the implications for business management structure.

## Introduction

The Social Brain Hypothesis suggests that there is a "natural" size of group for humans of around 150 [1,2]. There is considerable evidence that groupings of this size are characteristic of personal social networks both offline and online [3–7], as well as the size of villages and other natural groupings in both historical and contemporary small scale human societies [8,9]. Strictly speaking, although the value of 150 has a particular significance, the social brain relationship takes the form of a fractally structured series of group sizes that have the approximate sequence 5–15–50–150–500–1500 [10–11], a pattern also found (with the same numbers) in primates [12]. These numbers seem to act as natural attractors, and hence are more common as group sizes.

In humans, this sequence of numbers has been reported from a variety of social contexts, including patterns of interaction in several different kinds of social media [6,13–14], cellphone calling networks [15], personal social networks in contemporary societies [16], the structure of modern armies [17], the size distribution of residential campsites [18], alliance formation in online gaming worlds [19] and even science co-authorship networks [20]. Dunbar & Sosis [21] analysed both the size distributions and the size at community fission in several historical and

**Data Availability Statement:** All relevant data are within the Supporting Information file.

**Funding:** Tacit London Ltd is a company wholly owned by Emily Webber. Tacit did not have any role in the study design, data collection and analysis, decision to publish, or preparation of the

manuscript. The specific roles of the authors are articulated in the 'author contributions' section.

**Competing interests:** Tacit London Ltd is a company wholly owned by Emily Webber. This does not alter our adherence to PLOS ONE policies on sharing data and materials.

contemporary utopian communities and found that the numbers 50, 150 and 500 seem both to be more common than would be expected and allow communities to maximise longevity. There appears to be something about these particular numbers that makes them especially stable.

The value of 150 as a natural grouping size seems to be dictated by cognitive processing capacities associated with brain size, since this value is predicted by a robust relationship between group size and neocortex volume across primates (the Social Brain Hypothesis) [22,23]. There is now considerable evidence from neuroimaging experiments that there is a correlation between the size of friendship groups (egocentric social networks) measured in a variety of ways and the volumes of key brain regimes in the frontal, temporal and parietal lobes and the limbic system [24–35]. Similar results have also now been reported for primates [36,37]. These cortical regions are explicitly associated with social skills such as mentalising or mindreading [38–43].

Within this framework, however, it is the way we distribute our available social time that is largely responsible for the layered structure of these networks, both in humans [44,45] and animals [46,47]. This is a direct consequence of the fact that social time is limited: not only is the length of the active day constrained, but we have other important things to do in it [1,48–50]. The decisions we make about whom to prioritize, and how much time to invest in each of them in relation to the benefits that they provide us [51], create the layers when we try to optimize the number of individuals in each benefit category as a function of the respective costs and benefits [45,52].

There is no obvious explanation as to why viewing social networks from above (how individuals are distributed in space to create social groups) or from below (in the form of egocentric networks) should yield exactly the same fractal pattern with groupings and layers of identical sizes defined by the same frequencies of interaction. However, the pattern is widely observed in both humans and other primates and appears to be very robust. One possible reason is that egocentric networks map onto community structures because both have a modular structure derivative of the fact that our inner core groups (probably those associated with the 15-layer [46]) have a discrete size and higher order groupings are made up by bolting together other similar groupings connected by weaker bridging ties.

Given the pervasiveness of this pattern, we might expect it to apply in the context of business and administrative structures. So far, however, no studies have examined the structure of organisations in this light. We here analyse a sample of *communities of practice* (COP) as a particular example of this kind of business-world organisation. Communities of practice are groups of people that share a craft or profession [53–55]. Such communities exist to share knowledge and expertise, and can develop naturally out of interactions among practitioners or be created deliberately. Members do not have to be co-located, especially since the internet now allows geographically dispersed individuals to interact more easily than would previously have been the case [56]. As communities of shared expertise, COPs are not new: in many ways, they are a modern manifestation of medieval guilds without the regulatory component. Wenger et al. [57] offer a number of specific recommendations that are likely to maximise the success of a COP. These include allowing the community to evolve organically, allowing opportunities for dialogue, focusing on the value/purpose of the community, nurturing a regular rhythm for interaction within the community. In many ways, these mirror the natural processes that appear to be important in maintaining the cohesion of more conventional social communities [8,58].

We examine a sample of COPs drawn from a convenience sample. Respondents were asked to provide only information on the size of their COP and the type of activity involved. We use standard clustering methods to search for an optimal partition into sub-distributions in order

to identify natural clusters within the data. Since frequency of interaction is usually inversely related to grouping size in personal networks [3,43,59], we also ask whether frequency of meetings is inversely correlated with COP size, reflecting the difficulty of coordinating meetings among large numbers of people. A negative size-dependent relationship would provide some evidence that the time constraints that limit their size are the same as those that limit the size of natural social groupings. Because increasing size places additional stress on an organisation [60], we also ask whether management strategies change as COP size increases and, if they do, what form this takes. We are particularly interested in whether there is a transition from open, democratic management (all members help to organise and run the COP meetings in a collaborative way) to a more formal management structure (e.g. a management committee).

## Materials and methods

The data had been previously collected through a google form survey (*Communities of Practice at Work*). There were two main routes to recruiting participants for the survey: (1) through a relevant mailing list associated with EW's professional practice and (2) through followers of @ewebber https://twitter.com/ewebber on twitter. The mailing list consisted of 933 email addresses of people that had downloaded a copy of a community of practice maturity model at https://tacitlondon.com/community-of-practice-maturity-model between Feb 2016 and Feb 2019. The assumption was that these people had, at some point, been interested in the topic and so likely to have been part of an active COP. The twitter route was less targeted, but had the potential of initially reaching a network of ~9000 people around the world and many more following at least 61 retweets (shares) to an undetermined number of people. A total of 6 public tweets were sent at different times of day to maximise responses. These routes inevitably mean that many of the respondents were likely to occupy digital/IT industry roles within government, finance, retail and education sectors. However, this echoes the rise in communities of practice as part of digital/IT organisation structures. Most respondents are likely to have been based in the UK, although this cannot be confirmed as it was not a question on the survey.

The survey screened for people that were part of active COPs in the workplace. Community of practice was defined as: *Communities of practice (sometimes also called guilds, chapters, networks, communities of interest or clans) are groups of people who share a practice and interact regularly to support each other and grow their practice*. And relevant participants were defined as: *This survey is for people who are members of one or more communities of practice that support their work (either in the workplace or across workplaces). Please fill in one form for each community you are a member of*. In all, 111 people completed the survey and met the criterion for being a member of a COP. They reported being members of 130 different COPs. Our unit of analysis is the COP, not the individual respondent.

To determine how many people were in each active community, we asked: *Roughly how many members are there in the community*? A second question asked how many leaders there were (i.e. who were responsible for coordinating the group, setting its agenda and managing its meetings). A third question asked how often the community met (either all together or as a subset of members). The response field was free text. The responses were converted to a standardised numerical format (average number of weeks between meetings). Two final free text questions asked our informants to identify (1) what they most appreciated about their COP and (2) the major challenge or block to the effective functioning of the COP.

The data are available in the online S1 Data

To determine whether the data form a single homogenous dataset, we first compare the distribution of COP sizes against a normal, an exponential and a logarithmic distribution using a Kolmogorov-Smirnov one sample test. If none of these fit the data, it is likely that this is

because the data consist of a set of overlying distributions (e.g. a set of normal distributions with separate peaks whose tails overlap). If so, this should be apparent as a series of distinct breaks in slope when the data are plotted as a $log_{10}$-transformed cumulative distribution. The number of such breaks indicates the number of distinct clusters. We then use *k-means* cluster analysis in SPSS v.23 to search for the optimal distribution of datapoints within clusters so as to determine cluster mean values. We ran the analysis for $2 \le k \le 10$. In the limit, such analyses will always yield a perfect fit when the number of clusters equals the number of datapoints. The optimal number of clusters is usually taken to be that which optimises the goodness of fit, subject to the constraint that no cluster should contain too small a number of datapoints since this is not considered ideal. Coulson [61], for example, suggests a goodness of fit of around 85% as a reasonable trade off (for further discussion, see also [18]). As a first step, we plot the *F*-ratio for each analysis (as an index of goodness-of-fit) as a function of *k* and search for the value of *k* that maximises *F*.

The optimal number of clusters should be reflected in low rates of overlap in data allocations to adjacent clusters. To determine whether the F-ratio analysis yields an optimal partitioning of the data, we used silhouette analysis [62] since this is generally considered the most appropriate method. The silhouette statistic varies between -1 (complete overlap) and +1 (no overlap). Negative values imply that clusters are not well differentiated; silhouette values >0.3 indicate satisfactory resolution. Ideally, cluster means should remain stable as *k* increases: new clusters should arise by partitioning a single cluster rather than by reassigning clusters completely anew. To determine cluster stability, we counted the proportion of clusters whose mean values differed between *k = x* and *k = x+1* by more than the first decimal place. We plotted the resulting values against *k = x* in order to identify values of *k* where cluster mean value remains stable.

Finally, we seek to identify whether the observed mean cluster values correspond to the values identified in huntergatherer layer sizes. To do this, we follow [3,21] and determine whether individual cluster mean values at optimal *k* differ significantly from each of the observed mean values for layer sizes observed in human huntergatherer societies [63]; since the smallest formal grouping layer in these societies is ~50 in size, we used the values for the two innermost layers (those at ~5 and ~15) from egocentric networks [3]. We use the respective standard errors on the huntergatherer and network data to calculate *t* for each pairwise comparison, and seek the layer from which each cluster mean is least significantly different.

The data in this study were deemed to be third-party data that do not contain data from individual participants, and not subject to ethics review.

## Results

Fig 1 plots the distribution of COP group sizes in the sample. We first determine whether the distribution forms a simple univariate normal distribution. It does not (Kolmogorov-Smirnov one sample test with a mean of 180.0±493.8SD: p<0.001). An exponential distribution does not provide a better fit (p<0.001); nor does a $log_{10}$-transformation normalise the data (p<0.001). This likely means that the distribution is multimodal, formed from a number of separate distributions overlying each other. To test for this, we first plot the cumulative distribution on a $log_{10}$ scale (Fig 2A). Multimodal distributions can be detected by breaks in the slope of the distribution. The distribution suggests that there may be as many as six slope changes, all of which occur close to the nominal values of 5, 15, 50, 150, 500 and 1500 (identified by the dashed lines).

To determine the optimal number of such distributions, we use a *k*-mean cluster analysis, with *k* allowed to vary across the range $2 \le k \le 10$. Fig 2B plots the *F* ratio (as an index of goodness of fit) for successive values of *k*. The "elbow" in Fig 2B identifies *k = 6* as the optimal

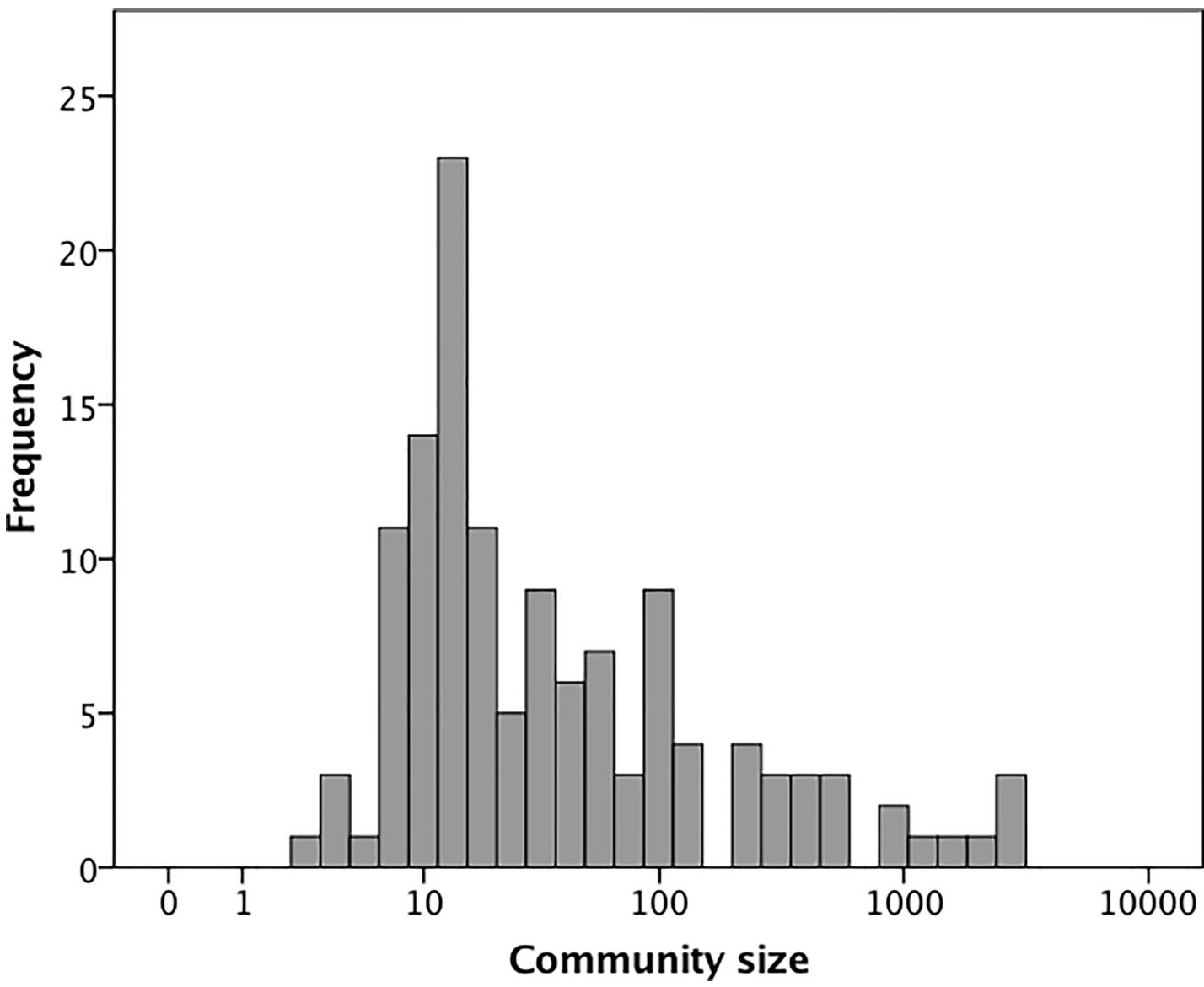

**Fig 1. Distribution of Community of Practice (COP) sizes.** The X-axis is $\log_{10}$-transformed for illustrative convenience.

number of clusters. We ran two separate tests to check this: cluster means should be stable above this value and a six cluster solution should yield a satisfactory silhouette pattern. Fig 3A plots the proportion of cluster means that do not change in value between $k = x$ and $k = x+1$. There appears to be a striking phase transition at $k = 6$: cluster means change a great deal in value below $k = 6$, but above it they change hardly at all. Moreover, the number of clusters with fewer than five members increases rapidly after $k = 6$, which undesirable. Second, we examined the silhouette pattern for the 6-cluster solution to determine whether the clusters are well separated. The mean silhouette value is 0.606, with all individual values >0.15, and a distribution that is significantly more positive than 0 (Fig 3B: one-sample Wilcoxon test with $H_0 = 0$: p<0.001). This is a very acceptable level of clustering. Increasing the number of clusters beyond 6 does not improve silhouette value (in fact, the mean varies only between 0.56 and 0.63 across the range of $k$) suggesting that the breaks between clusters are quite stable.

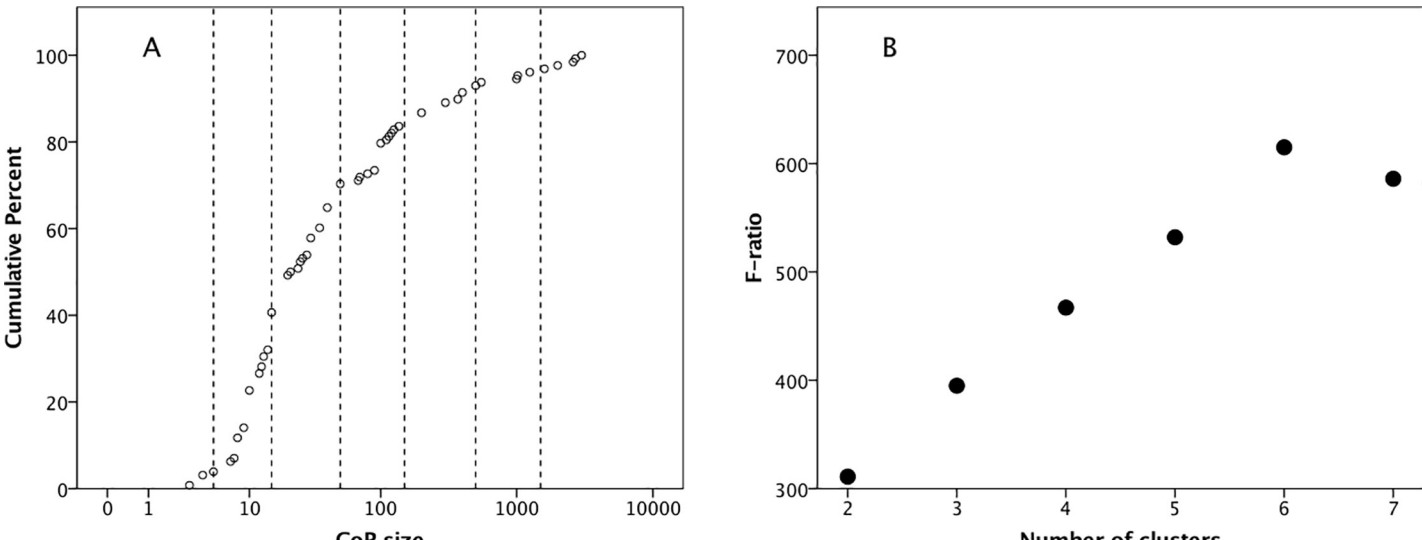

**Fig 2.** (a) Cumulative distribution of size of COPs. Vertical lines demarcate (L to R) 5, 15, 50, 150, 500 and 1500. Distinctive breaks in slope are evident in the vicinity of these lines. (b) Goodness of fit (indexed as *F*-ratio) for *k*-means cluster analysis of Community of Practice group sizes as a function of number of clusters.

The cluster means identified by this analysis are: 4.0, 11.0, 30.2, 112.2, 389.0, 1737.8, with a mean scaling ratio of 3.4. We next ask whether each of these values fit the mean layer values found for huntergatherer social organisation. To do this, we compare each observed cluster mean value with each of the huntergatherer layer values and calculate a *t* value based on the appropriate standard error for the huntergatherer data. The results are given in Table 1. For each case, the observed value differs significantly in size from all but one of the corresponding layers in hunter-gatherer communities. In two cases, an adjacent value is also not significant (but always with a markedly higher value of *t* that approaches statistical significance). To give a sense of the scale of the fit, the mean value of the *t*-statistic for the putative corresponding

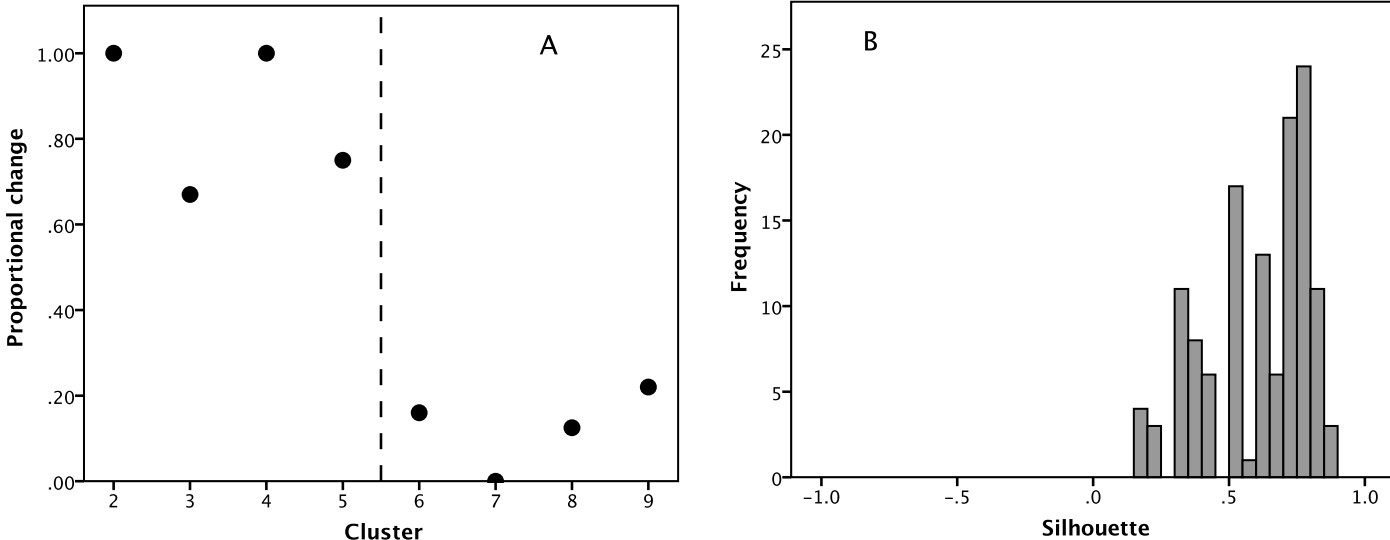

**Fig 3.** (a) Proportion of clusters whose mean size changes by more than the first decimal place between the *k = x* and *k = x+1* cluster solution. (b) Distribution of silhouette values for a *k = 6* solution.

**Table 1. Student's *t* comparing observed cluster means against theoretical values derived from previous studies.**

| Observed | t against predicted values | | | | | |
|---|---|---|---|---|---|---|
| Means | 3.8±2.39† | 11.3±6.19† | 42.8±18.0‡ | 127.0±43.8‡ | 566.6±166.2‡ | 1727.9±620.6‡ |
| 4.0 | *0.08*¶ | -1.18 | -2.16* | -2.81* | -3.39* | -2.78* |
| 11.0 | 3.01* | *-0.05*¶ | 1.76 | -2.65* | -3.34* | -2.78* |
| 30.2 | 11.05* | 3.05* | *0.70* | -2.21* | -3.23* | -2.74* |
| 112.2 | 45.27* | 16.30* | 3.86* | *-0.34* | -2.73* | -2.60* |
| 389.0 | 161.17* | 61.02* | 19.23* | 5.98* | *-1.07* | -2.16* |
| 1737.8 | 725.52* | 278.92* | 94.17* | 36.78* | 7.05* | *0.02*¶ |

† values for egocentric social networks from [3]

‡ values for hunter-gatherer societies from [63]

* For N = 128, p<0.05 when *t*<1.98; best fit value in each case is indicated in **bold italics**.

¶ When *t*<0.062, p>0.95 2-tailed, indicating significant similarity.

layers (the cells on the main diagonal) is t = 0.37±0.43SD (N = 6) versus t = 50.40±140.32 (N = 30) for the off-diagonal cells. The difference is highly significant (Mann-Whitney test: p<0.001).

Each COP was managed either by an individual or a small subset of the membership or by all the members acting as a collective (e.g. by taking it in turns to organise meetings). Fig 4 plots the number of leaders (as defined by the informant) against community size. The mean number of leaders is 4.09±5.72SD (indicated by the horizontal dotted line). The pattern on the left hand side of the graph suggests a divergence between those COPs that run themselves as a democratic collective (the line of datapoints to the left of the dashed vertical line that rise in line with COP size) and those that have some kind of management structure with one or more formal leaders (the line of datapoints along the base below the horizontal dotted line). The attempt to work with an effectively leaderless system (i.e. one in which every member can initiate and manage particular events, even if there is one person who acts as coordinator) seems to reach its limit at COPs of ~40 members (the dashed vertical line): after that, there is a phase transition (demarcated by the vertical dashed line) marking a sudden switch from the possibility of democracy to management by a small team. A plot of the distribution of leadership team size (Fig 5) suggests a bimodal distribution involving either a single individual (occasionally two people) or a group of 5–10 individuals. The ratio of these two options does not appear to correlate with COP size.

Most COP s (30%) met weekly either in person or online. However, the distribution was skewed with the average interval between successive meetings being 6.3±12.6D weeks. As might be expected given the need to coordinate large numbers of individuals, there is a significant positive relationship between the inter-meeting interval and COP size for the six clusters (Fig 6; regression on raw data: $r^2 = 0.262$, $F_{1,123} = 43.6$, p<0.0001). For the nominal grouping levels, the mean intervals between meetings are 12.6 days for groups of 5, 23.9 days for groups of 15, 25.0 days for groups of 50, 46.3 days for groups of 150, 64.2 days for groups of 500 and 245.9 days for groups of 1500. These values roughly approximate the equivalent contact frequencies between individuals that we find in both face-to-face and online communities: for the first four layers, these are once every 6.4 days, 17.2 days, 72.5 days and 143.7 days, respectively [14]. Further evidence of the constraints imposed by time come from the open text answers to our question on the challenges that informants felt prevented COPs from fulfilling their potential. Of the 130 cases, 40 (30.5%) listed time to attend meetings as the most important problem, and a further 34 (26%) identified other members' failure to be involved in organising meetings. In other, almost two-thirds of COPs were hampered by their members' time constraints.

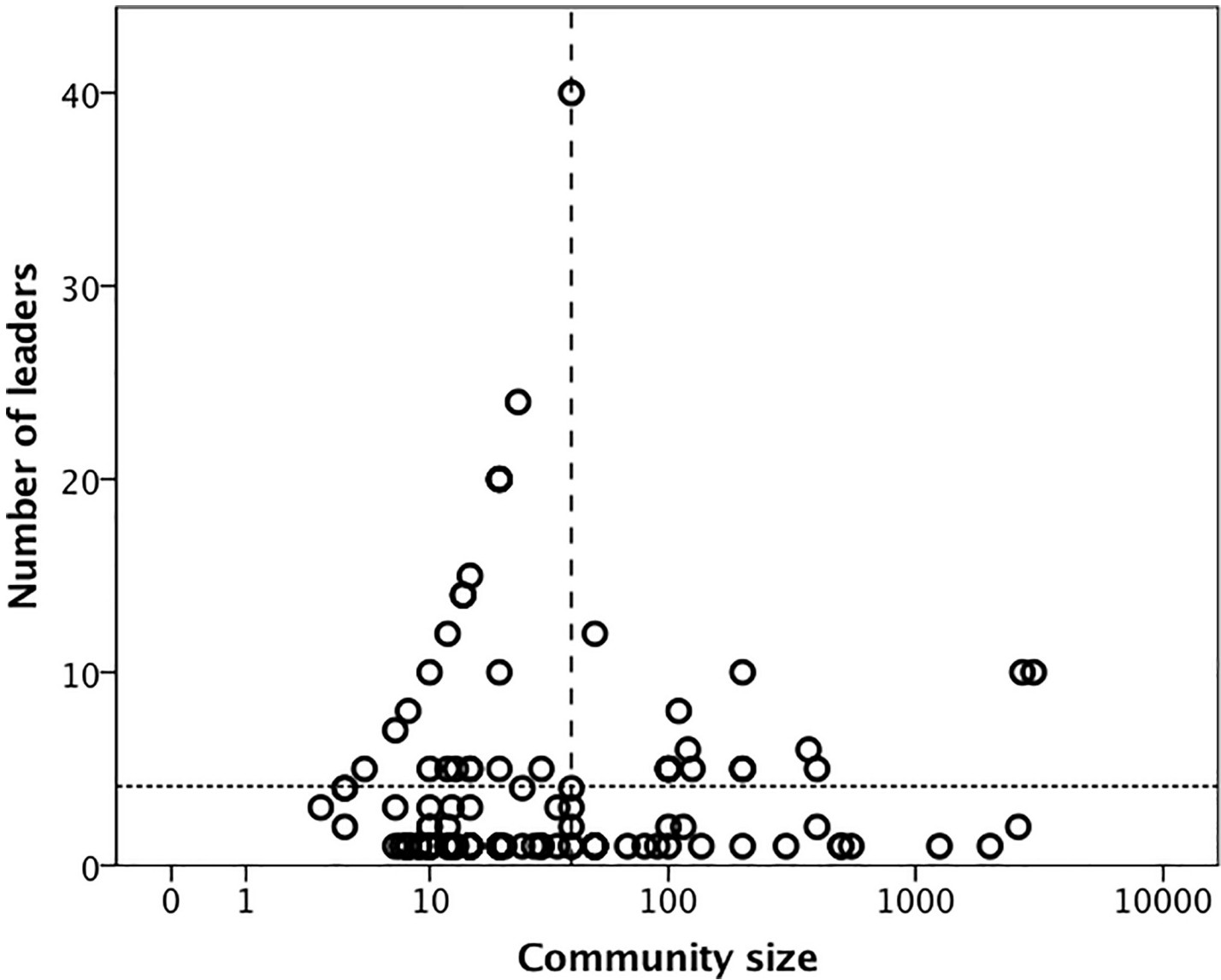

**Fig 4. Number of leaders in the community plotted against community size.** Horizontal dotted line denotes overall mean (4.09 leaders). Vertical dashed line denotes the apparent upper limit for democratic management of communities. The X-axis is $\log_{10}$-transformed for illustrative convenience.

From the free-text answers to the question "What do you most appreciate about your COP?", one response, which we label a sense of camaraderie, was especially common. Many of the informants explicitly used this term, but we included comments that mentioned trust, fun, engagement or supportiveness. Significantly more respondents in small COPs (<40 in size) stated that a sense of camaraderie was the single most important factor about their COP compared to those in large (>40) COPs (36.5% of 85 cases vs 17.8% of 45 cases, respectively: Fisher Exact test, p = 0.029).

## Discussion

Communities of Practice seem to have a similar fractal distribution to natural human communities, as reflected in both the structure of personal social networks and the distribution of

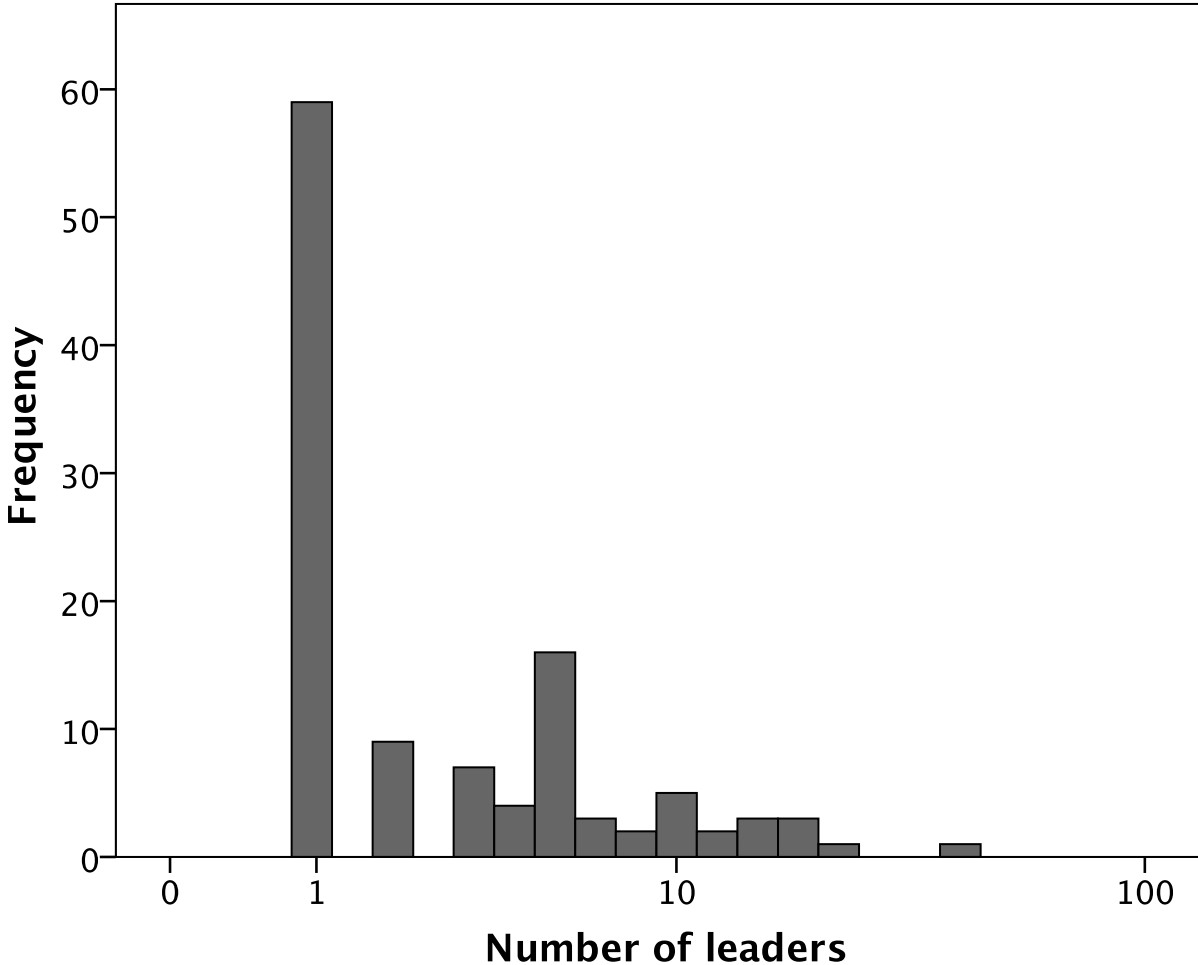

**Fig 5. Frequency distribution for size of leadership group.**

nested hunter-gatherer social groupings. Business communities are typically more functionally focussed than conventional social communities, and this may, of course, make them more transient [64]. In this respect, they may resemble the more transient networks characteristic of twitter [4,13] and, perhaps, academic networks [5,20]. Although hunter-gatherer communities clearly have economic functions, nonetheless they are also social communities based on historically deep personal relationships of kinship and friendship between the individuals and families that, in most cases, last a lifetime (an individual's tribal membership, for example, does not change during their life).

Business communities are necessarily task-oriented and may dissolve quite quickly once the task has been accomplished [64], much as alliances in the online gaming world do [19]. Where COPs form professional associations, they may, of course, be longer lasting, although even here there may be considerable turnover ("churn") in membership over time. Most millennials, for example, only expect to stay in a job for 4.4 years on average [65], and this may exacerbate the rate of membership churn even in those cases where a COP has an extended lifespan. The fact that COPs have a similar fractal structure to natural human groupings suggests that the limits on group size are set by the same kinds of constraints as determine the size

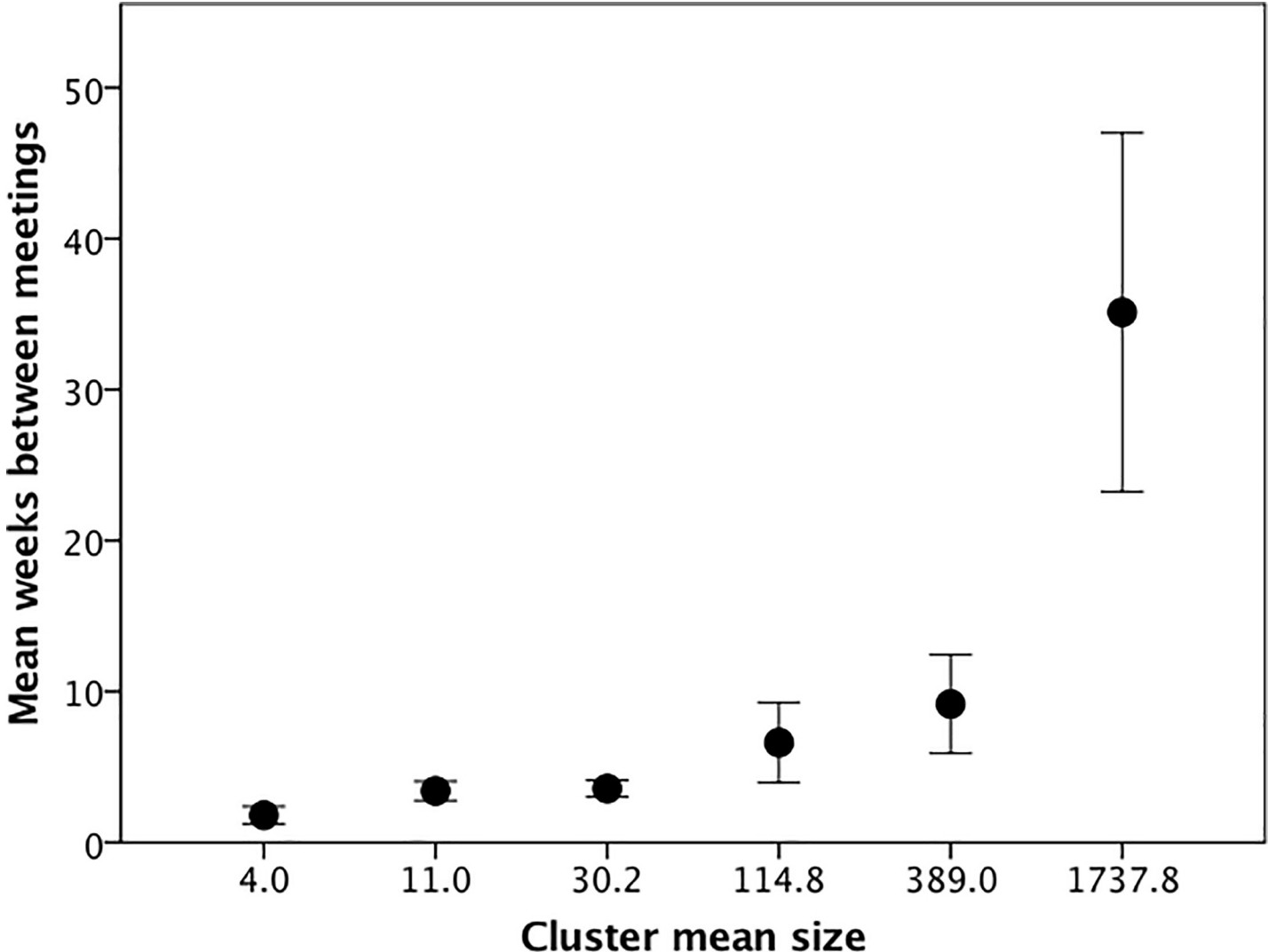

**Fig 6. Mean (±1 se) time between meetings as a function of cluster.** Note that, in a small number of cases, frequency of meetings was not specified.

of natural social human groups. These constraints normally relate to a combination of time [44,45,59] and mental processing capacity [26,27,31,34]. The neural demands of managing social relationships has, for example, been shown to be considerably greater than managing purely factual information [41]. This suggests that the cognitive demands of managing large numbers of relationships probably influences the size of COPs in much the same way as they do the size of human social groups.

Fig 4 suggests that it is possible for communities to be run without having a formal leadership structure up to a limiting size of about 40 individuals, but that above this size community coherence requires the emergence of a more formal management structure (e.g. a management committee or some kind of acknowledged leadership). This may explain why huntergatherer bands (or camp groups) (which vary between about 35–50 in size depending on latitude [66]) are able to function effectively as egalitarian, democratic institutions, whereas forms of leadership tend to emerge in larger scale communities (e.g. tribes). Johnson [60] has suggested that natural human groups face stresses that grow exponentially as group size increases. This sets

an upper limit on group size that can only be breached if the group becomes substructured. This substructuring, he argued, emerges at a group size of about 40. Dávid-Barrett & Dunbar [67] showed that small elites emerge naturally in communities whose functional objective is group coordination (as opposed to cooperation on a task) and, when they do, the presence of a managing elite significantly increases the efficiency with which the community functions (at least in terms of information flow). This naturally gives rise to a layered structure to such communities.

The layered structure in human social networks is thought to reflect mainly the time costs of maintaining relationships [44,45,52]. In the present case, this is reflected in the negative relationship between COP size and the interval between meetings (Fig 5). That time may be a constraint is hinted at by the fact that, in our sample, time commitments were identified as the major challenge limiting COPs from achieving their full potential. Members have to be willing to sacrifice time devoted to other activities (or, in a social context, other individuals) in order to attend meetings. The more frequent these meetings are, the greater that commitment to the group will have to be. An important component of that commitment will be the sense of obligation to other COP members, and this sense of obligation is likely to be greater and more personalised when the group size is small (creating a greater pressure to attend). In this case, the group will need to meet frequently in order to generate a sufficiently strong sense of personal obligation.

The phase shift in COP management structure (Fig 4) provides us with some valuable insights into why substructuring might have to occur in both social and business organisation: it rapidly becomes impossible to manage more than 40–50 individuals in a strictly democratic face-to-face fashion. Some structural mechanism is needed to allow face-to-face interactions to function effectively in a way that facilitates organisational management. Hutterite farming communities (known as "colonies") provide a relevant, if unusual, example of this. Their communities are run on strictly democratic lines despite the fact that they typically contain 100–150 individuals of all ages [21]. Fig 4 suggests that this ought to be beyond their capacity to manage democratically. Hutterites solve the problem by a form of covert structuring–management decisions are made only by the adult males of the community (typically around 20–40 individuals) who constitute a small enough group to function along strict democratic lines.

Our findings have implications for organisations wanting to benefit from COPs at a number of levels. These relate to community size, psychological intimacy, formal management mechanisms, and the availability of time. It goes without saying, perhaps, senior managers need to adopt a supportive attitude. Indeed, lack of such support was mentioned by a number of respondents as the main constraint on the success of COPs.

It is clear that size matters and influences the benefits derived from a community of practice. Smaller COPs allow for greater psychological intimacy and camaraderie, and this will have an impact on feelings of belonging and support. This may well impact on staff retention and recruitment. Smaller communities have the ability to meet more frequently, which means increasing the camaraderie further. These communities create stronger bonds between individuals and lean towards becoming social networks because of this. The bonds involved in such a network can create enough commitment to maintain the community without the need for organisational management intervention. These smaller more intimate groups are able to have a diverse set of activities that are possible because of their smaller nature. They include, sharing, learning, face to face discussions, "brown bag" sessions, social events and collaborative projects (e.g. creating best practice guidelines, setting standards, producing training materials). Larger communities necessarily require a different approach. Due to the nature of meeting frequency, they may need to be more formally managed, either through time allocated to existing community members to form a committee or additional support staff. A COP moving from a

smaller size (less than 40) to a larger size will need a transition to help make that happen. In addition, larger groups are by their nature restricted in the activities that they can carry out, but may develop additional functions such as training, showcases, career opportunities, newsletters, conferences and discussion forums. Very large communities can be seen as a network, where a core group broadcasts information to a wider group. Because of this, there might be the need to create subsets in larger communities to carry out more focused activities and tasks

All COPs exist only because one or more individuals are willing, and have the capacity, to take on the task of organising them. It was clear from the comments to the free text questions that these individuals find this a burden (a quarter listed failure by other members to contribute as a major limitation to the success of the COP). The phase shift in management structure at COPs of around 40 in size raises the more general point that a large COP or voluntary associations will only survive if there is a pool of people willing and able to undertake the task of running it (i.e. by joining its management committee). Organisations can support this by freeing up people's time to manage the community, actively encouraging or rewarding organisational and participation through support, sponsorship or incentives, and by dedicating individuals to take on support roles for the COPs.

The main lesson would seem to be how to maintain a balance between an open, democratic organisation in which all members feel they have a stake and effective top-down management once an organisation (or a department within a larger organisation) exceeds ~50 individuals in size. It seems that this circle cannot easily be squared without developing a formal management structure.

## Supporting information

**S1 Data.**
(CSV)

## Author Contributions

**Conceptualization:** Emily Webber.

**Data curation:** Emily Webber.

**Formal analysis:** Robin Dunbar.

**Methodology:** Emily Webber.

**Writing – original draft:** Robin Dunbar.

**Writing – review & editing:** Emily Webber.

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
