## [Decision Letter · Decision Letter 0]

3 Mar 2020

PONE-D-19-29833

The Fractal Structure of Communities of Practice

PLOS ONE

Dear Dr. Dunbar,

Thank you for submitting your manuscript to PLOS ONE. After careful consideration, we feel that it has merit but does not fully meet PLOS ONE’s publication criteria as it currently stands. Therefore, we invite you to submit a revised version of the manuscript that addresses the points raised during the review process.

We would appreciate receiving your revised manuscript by Apr 10 2020 11:59PM. To enhance the reproducibility of your results, we recommend that if applicable you deposit your laboratory protocols in protocols.io, where a protocol can be assigned its own identifier (DOI) such that it can be cited independently in the future. For instructions see: http://journals.plos.org/plosone/s/submission-guidelines#loc-laboratory-protocols

We look forward to receiving your revised manuscript.

Kind regards,

Jichang Zhao, Ph.D.

Academic Editor

PLOS ONE

Journal Requirements:

2. Please consider changing the title so as to meet our title format requirement (https://journals.plos.org/plosone/s/submission-guidelines). In particular, the title should be "Specific, descriptive, concise, and comprehensible to readers outside the field" and in this case it is not informative and specific about your study's scope and methodology.

4. Thank you for stating the following in the Financial Disclosure section:

The authors received no specific funding for this work.

We note that one or more of the authors are employed by a commercial company: Tacit London Ltd.

Reviewers' comments:

Reviewer's Responses to Questions

**Comments to the Author**

1. Is the manuscript technically sound, and do the data support the conclusions?

Reviewer #1: Yes

Reviewer #2: Partly

2. Has the statistical analysis been performed appropriately and rigorously? 

Reviewer #1: Yes

Reviewer #2: Yes

3. Have the authors made all data underlying the findings in their manuscript fully available?

Reviewer #1: Yes

Reviewer #2: Yes

4. Is the manuscript presented in an intelligible fashion and written in standard English?

Reviewer #1: Yes

Reviewer #2: Yes

5. Review Comments to the Author

Reviewer #1: In this manuscript, authors analyzed the fractal structure of communities of practice and revealed the relation between the size of CoP and the administrative structures. This work is of great importance and interests. And the manuscript can be improved if the following comments and suggestions can be considered in the revision.

1 In the introduction, it is better to provide clear and sufficient contribution to readers.

2 Some writing errors should be corrected. For example, in page 6, “SPPS” should be “SPSS”.

3 More clarity is required on what exactly the F-ratio and Silhouette index.

4 In figure 4, there is a phase transition from the possibility of democracy to management by a small team of either 1-2 or 5-10 individuals. Why there are two types of management team size? Is there any intuitive explanation?

5 While the discussion section seems adequate, I suggest to add more to the application of results in organization management. I suppose the analysis in the context of business and administrative structures is a major contribution, some implications for business application would be really beneficial.

Reviewer #2: The paper analyses the size of community of practises. The main goal is to understand if natural groupings exist, giving rise to regular structures in social groupings, as it was demonstrated consistently in the literature both for offline and online communication means and ways of meeting.

To this end, authors collect a survey of around 100 individuals, who self report the sizes of the CoP they belong to. The paper provides (i) a cluster analysis to identify common sizes of communities; (ii) an analysis of the number of leaders that emerge as a function of the community size.

The paper derives the following main results: (i) well-formed clusters exist in group sizes. Specifically, 6 clusters can be found, and the sizes of typical human group sizes previously found in the literature are good fits for those sizes. (ii) communities cannot grow using an egalitarian way of management (i.e., without a management structure) above around 40 members, which is a size similar to those found in “non-layered” hunter-gatherer societies.

The paper provides interesting results on a less explored type of human social networks, with respect to existing literature. To the best of the reviewer’s knowledge, the structure of CoP was never analysed before.

The analysis of the collected result is methodologically sound, and the provided results in general back up well the claims about the existence of typical sizes for CoP communities, and the “scalability” properties of managing such groups.

One point, in this reviewer’s opinion, needs much better discussion, and maybe to gather additional data. Specifically, in the existing literature, the grouping of personal networks are normally considered from the ego standpoint, not from the entire community standpoint. Even more importantly, the sizes of the clusters typically observed in literature refer to individual relationships between an ego and their social ties. The same holds also for the frequency of interactions. Conceptually, making the analogy between groups in individual ego networks and entire communities is not straightforward, and would need to be better justified.

Another point to be at least discussed in the paper is whether there exists an internal structure inside the observed communities. For example, do the large communities present internal clusters of individuals interacting at higher frequencies? Can layered structures be observed (as it is the case in most of the related literature authors consider)?

Finally, in this reviewer’s opinion, two aspects of the methodology should be described in more detail. Specifically, authors should explain more in detail the methodology behind the analysis of Figure 2c and Table 1, which are both described in a rather cursory way.

6. PLOS authors have the option to publish the peer review history of their article (what does this mean?). If published, this will include your full peer review and any attached files.

Reviewer #1: No

Reviewer #2: No

---

## [Author Response · Author response to Decision Letter 0]

31 Mar 2020

Journal Requirements:

DONE 

2. Please consider changing the title so as to meet our title format requirement (https://journals.plos.org/plosone/s/submission-guidelines). In particular, the title should be "Specific, descriptive, concise, and comprehensible to readers outside the field" and in this case it is not informative and specific about your study's scope and methodology.

Title amended

It already stated in the Methods section at the data were provided in the Supplementary material. Both reviewers noted this.

This is now done.

4. Thank you for stating the following in the Financial Disclosure section:

The authors received no specific funding for this work.

We note that one or more of the authors are employed by a commercial company: Tacit London Ltd.

Tacit is the wholly owned company of the first author: it does not employ her. The Acknowledgments have been revised as requested.

This is now done.

PLEASE NOTE: IT SEEMS NOT TO BE POSSIBLE TO CHANGE THE FUNDING STATEMENT ON THE SYSTEM!

Now updated

Reviewers' comments:

Comments to the Author

Reviewer #1: 

1) In the introduction, it is better to provide clear and sufficient contribution to readers.

We have added additional material to the Introduction to address this point.

2) Some writing errors should be corrected. For example, in page 6, “SPPS” should be “SPSS”.

corrected

3) More clarity is required on what exactly the F-ratio and Silhouette index.

detail added to Methods section

4) In figure 4, there is a phase transition from the possibility of democracy to management by a small team of either 1-2 or 5-10 individuals. Why there are two types of management team size? Is there any intuitive explanation?

Thank you for the suggestion. We have now added some comments on this to the Discussion.

5) While the discussion section seems adequate, I suggest to add more to the application of results in organization management. I suppose the analysis in the context of business and administrative structures is a major contribution, some implications for business application would be really beneficial.

We have added some additional material to the Discussion to address this interesting point.

Reviewer #2: 

(1) One point, in this reviewer’s opinion, needs much better discussion, and maybe to gather additional data. Specifically, in the existing literature, the grouping of personal networks are normally considered from the ego standpoint, not from the entire community standpoint. Even more importantly, the sizes of the clusters typically observed in literature refer to individual relationships between an ego and their social ties. The same holds also for the frequency of interactions. Conceptually, making the analogy between groups in individual ego networks and entire communities is not straightforward, and would need to be better justified.

In fact, the literature does show that the layered structure of ego networks and of social communities is identical. However, we have added further comment on this to the Introduction to make the justification clearer.

(2) Another point to be at least discussed in the paper is whether there exists an internal structure inside the observed communities. For example, do the large communities present internal clusters of individuals interacting at higher frequencies? Can layered structures be observed (as it is the case in most of the related literature authors consider)?

A good point. In fact, we have already commented on this point in respect of the need for a two-tier structure (organising committee + members in groups with >50 vs members only in small groups). We have added additional comments to the Discussion.

(3) Finally, in this reviewer’s opinion, two aspects of the methodology should be described in more detail. Specifically, authors should explain more in detail the methodology behind the analysis of Figure 2c and Table 1, which are both described in a rather cursory way.

Detail now added to Methods.

---

## [Editor Report · Decision Letter 1]

10 Apr 2020

The fractal structure of communities of practice: Implications for business organization

PONE-D-19-29833R1

Dear Dr. Dunbar,

We are pleased to inform you that your manuscript has been judged scientifically suitable for publication and will be formally accepted for publication once it complies with all outstanding technical requirements.

With kind regards,

Jichang Zhao, Ph.D.

Academic Editor

PLOS ONE
---

## [Editor Report · Acceptance letter]

15 Apr 2020

PONE-D-19-29833R1 

The fractal structure of communities of practice: Implications for business organization 

Dear Dr. Dunbar:

I am pleased to inform you that your manuscript has been deemed suitable for publication in PLOS ONE. Congratulations! Your manuscript is now with our production department. 

With kind regards,

on behalf of

Professor Jichang Zhao 

Academic Editor

PLOS ONE